# Co-teaching: Robust Training of Deep Neural Networks with Extremely Noisy Labels

**Bo Han**[*1,2], **Quanming Yao**[*3], **Xingrui Yu**[1], **Gang Niu**[2],
**Miao Xu**[2], **Weihua Hu**[4], **Ivor W. Tsang**[1], **Masashi Sugiyama**[2,5]

[1]Centre for Artificial Intelligence, University of Technology Sydney;
[2]RIKEN; [3]4Paradigm Inc.; [4]Stanford University; [5]University of Tokyo

## Abstract

Deep learning with noisy labels is practically challenging, as the capacity of deep models is so high that they can totally memorize these noisy labels sooner or later during training. Nonetheless, recent studies on the *memorization effects* of deep neural networks show that they would first memorize training data of clean labels and then those of noisy labels. Therefore in this paper, we propose a new deep learning paradigm called "*Co-teaching*" for combating with noisy labels. Namely, we train two deep neural networks simultaneously, and let them *teach each other* given every mini-batch: firstly, each network feeds forward all data and selects some data of possibly clean labels; secondly, two networks communicate with each other what data in this mini-batch should be used for training; finally, each network back propagates the data selected by its peer network and updates itself. Empirical results on noisy versions of *MNIST*, *CIFAR-10* and *CIFAR-100* demonstrate that Co-teaching is much superior to the state-of-the-art methods in the robustness of trained deep models.

## 1  Introduction

Learning from noisy labels can date back to three decades ago [1], and still keeps vibrant in recent years [13, 31]. Essentially, noisy labels are corrupted from ground-truth labels, and thus they inevitably degenerate the robustness of learned models, especially for deep neural networks [2, 45]. Unfortunately, noisy labels are ubiquitous in the real world. For instance, both online queries [4] and crowdsourcing [42, 44] yield a large number of noisy labels across the world everyday.

As deep neural networks have the high capacity to fit noisy labels [45], it is challenging to train deep networks robustly with noisy labels. Current methods focus on estimating the noise transition matrix. For example, on top of the softmax layer, Goldberger et al. [13] added an additional softmax layer to model the noise transition matrix. Patrini et al. [31] leveraged a two-step solution to estimating the noise transition matrix heuristically. However, the noise transition matrix is not easy to be estimated accurately, especially when the number of classes is large.

To be free of estimating the noise transition matrix, a promising direction focuses on training on selected samples [17, 26, 34]. These works try to select clean instances out of the noisy ones, and then use them to update the network. Intuitively, as the training data becomes less noisy, better performance can be obtained. Among those works, the representative methods are MentorNet [17] and Decoupling [26]. Specifically, MentorNet pre-trains an extra network, and then uses the extra network for selecting clean instances to guide the training. When the clean validation data is not available, MentorNet has to use a predefined curriculum (e.g., self-paced curriculum). Nevertheless, the idea of self-paced MentorNet is similar to the self-training approach [6], and it inherited the same inferiority of accumulated error caused by the sample-selection bias. Decoupling trains two networks

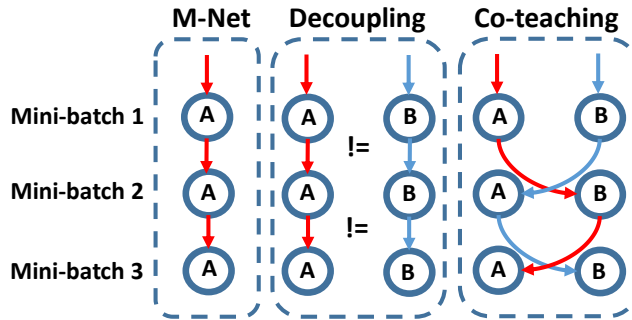

Figure 1: Comparison of error flow among MentorNet (M-Net) [17], Decoupling [26] and Co-teaching. Assume that the error flow comes from the biased selection of training instances, and error flow from network A or B is denoted by red arrows or blue arrows, respectively. **Left panel:** M-Net maintains only one network (A). **Middle panel:** Decoupling maintains two networks (A & B). The parameters of two networks are updated, when the predictions of them disagree (!=). **Right panel:** Co-teaching maintains two networks (A & B) simultaneously. In each mini-batch data, each network samples its small-loss instances as the useful knowledge, and teaches such useful instances to its peer network for the further training. Thus, the error flow in Co-teaching displays the zigzag shape.

simultaneously, and then updates models only using the instances that have different predictions from these two networks. Nonetheless, noisy labels are evenly spread across the whole space of examples. Thus, the disagreement area includes a number of noisy labels, where the Decoupling approach cannot handle noisy labels explicitly. Although MentorNet and Decoupling are representative approaches in this promising direction, there still exist the above discussed issues, which naturally motivates us to improve them in our research.

Meanwhile, an interesting observation for deep models is that they can memorize easy instances first, and gradually adapt to hard instances as training epochs become large [2]. When noisy labels exist, deep learning models will eventually memorize these wrongly given labels [45], which leads to the poor generalization performance. Besides, this phenomenon does not change with the choice of training optimizations (e.g., Adagrad [9] and Adam [18]) or network architectures (e.g., MLP [15], Alexnet [20] and Inception [37]) [17, 45].

In this paper, we propose a simple but effective learning paradigm called "*Co-teaching*", which allows us to train deep networks robustly even with extremely noisy labels (e.g., 45% of noisy labels occur in the fine-grained classification with multiple classes [8]). Our idea stems from the Co-training approach [5]. Similarly to Decoupling, our Co-teaching also maintains two networks simultaneously. That being said, it is worth noting that, in each mini-batch of data, each network views its small-loss instances (like self-paced MentorNet) as the useful knowledge, and teaches such useful instances to its peer network for updating the parameters. The intuition why Co-teaching can be more robust is briefly explained as follows. In Figure 1, assume that the error flow comes from the biased selection of training instances in the first mini-batch of data. In MentorNet or Decoupling, the error from one network will be directly transferred back to itself in the second mini-batch of data, and the error should be increasingly accumulated. However, in Co-teaching, since two networks have different learning abilities, they can filter different types of error introduced by noisy labels. In this exchange procedure, the error flows can be reduced by peer networks mutually. Moreover, we train deep networks using stochastic optimization with momentum, and nonlinear deep networks can memorize clean data first to become robust [2]. When the error from noisy data flows into the peer network, it will attenuate this error due to its robustness.

We conduct experiments on noisy versions of *MNIST*, *CIFAR-10* and *CIFAR-100* datasets. Empirical results demonstrate that, under extremely noisy circumstances (i.e., 45% of noisy labels), the robustness of deep learning models trained by the Co-teaching approach is much superior to state-of-the-art baselines. Under low-level noisy circumstances (i.e., 20% of noisy labels), the robustness of deep learning models trained by the Co-teaching approach is still superior to most baselines.

## 2 Related literature

**Statistical learning methods.** Statistical learning contributed a lot to the problem of noisy labels, especially in theoretical aspects. The approach can be categorized into three strands: surrogate loss,

**Algorithm 1** Co-teaching Algorithm.

1: **Input** $w_f$ and $w_g$, learning rate $\eta$, fixed $\tau$, epoch $T_k$ and $T_{\max}$, iteration $N_{\max}$;

**for** $T = 1, 2, \ldots, T_{\max}$ **do**

    2: **Shuffle** training set $\mathcal{D}$;                                                           //noisy dataset

    **for** $N = 1, \ldots, N_{\max}$ **do**

        3: **Fetch** mini-batch $\bar{\mathcal{D}}$ from $\mathcal{D}$;

        4: **Obtain** $\bar{\mathcal{D}}_f = \arg\min_{\mathcal{D}':|\mathcal{D}'| \geq R(T)|\bar{\mathcal{D}}|} \ell(f, \mathcal{D}')$;    //sample $R(T)$% small-loss instances

        5: **Obtain** $\bar{\mathcal{D}}_g = \arg\min_{\mathcal{D}':|\mathcal{D}'| \geq R(T)|\bar{\mathcal{D}}|} \ell(g, \mathcal{D}')$;    //sample $R(T)$% small-loss instances

        6: **Update** $w_f = w_f - \eta\nabla\ell(f, \bar{\mathcal{D}}_g)$;                            //update $w_f$ by $\bar{\mathcal{D}}_g$;

        7: **Update** $w_g = w_g - \eta\nabla\ell(g, \bar{\mathcal{D}}_f)$;                            //update $w_g$ by $\bar{\mathcal{D}}_f$;

    **end**

    8: **Update** $R(T) = 1 - \min\left\{\frac{T}{T_k}\tau, \tau\right\}$;

**end**

9: **Output** $w_f$ **and** $w_g$.

---

noise rate estimation and probabilistic modeling. For example, in the surrogate losses category, Natarajan et al. [30] proposed an unbiased estimator to provide the noise corrected loss approach. Masnadi-Shirazi et al. [27] presented a robust non-convex loss, which is the special case in a family of robust losses. In the noise rate estimation category, both Menon et al. [28] and Liu et al. [23] proposed a class-probability estimator using order statistics on the range of scores. Sanderson et al. [36] presented the same estimator using the slope of the ROC curve. In the probabilistic modeling category, Raykar et al. [32] proposed a two-coin model to handle noisy labels from multiple annotators. Yan et al. [42] extended this two-coin model by setting the dynamic flipping probability associated with instances.

**Other deep learning approaches.** In addition, there are some other deep learning solutions to deal with noisy labels [24, 41]. For example, Li et al. [22] proposed a unified framework to distill the knowledge from clean labels and knowledge graph, which can be exploited to learn a better model from noisy labels. Veit et al. [40] trained a label cleaning network by a small set of clean labels, and used this network to reduce the noise in large-scale noisy labels. Tanaka et al. [38] presented a joint optimization framework to learn parameters and estimate true labels simultaneously. Ren et al. [34] leveraged an additional validation set to adaptively assign weights to training examples in every iteration. Rodrigues et al. [35] added a crowd layer after the output layer for noisy labels from multiple annotators. However, all methods require either extra resources or more complex networks.

**Learning to teach methods.** Learning-to-teach is also a hot topic. Inspired by [16], these methods are made up by teacher and student networks. The duty of teacher network is to select more informative instances for better training of student networks. Recently, such idea is applied to learn a proper curriculum for the training data [10] and deal with multi-labels [14]. However, these works do not consider noisy labels, and MentorNet [17] introduced this idea into such area.

## 3 Co-teaching meets noisy supervision

Our idea is to train two deep networks simultaneously. As in Figure 1, in each mini-batch data, each network selects its small-loss instances as the useful knowledge, and teaches such useful instances to its peer network for the further training. Therefore, the proposed algorithm is named *Co-teaching* (Algorithm 1). As all deep learning training methods are based on stochastic gradient descent, our Co-teaching works in a mini-batch manner. Specifically, we maintain two networks $f$ (with parameter $w_f$) and $g$ (with parameter $w_g$). When a mini-batch $\bar{\mathcal{D}}$ is formed (step 3), we first let $f$ (resp. $g$) select a small proportion of instances in this mini-batch $\bar{\mathcal{D}}_f$ (resp. $\bar{\mathcal{D}}_g$) that have small training loss (steps 4 and 5). The number of instances is controlled by $R(T)$, and $f$ (resp. $g$) only selects $R(T)$ percentage of small-loss instances out of the mini-batch. Then, the selected instances are fed into its peer network as the useful knowledge for parameter updates (steps 6 and 7).

There are two important questions for designing above Algorithm 1:

    Q1. Why can sampling small-loss instances based on dynamic $R(T)$ help us find clean instances?

    Q2. Why do we need two networks and cross-update the parameters?

To answer the *first question*, we first need to clarify the connection between small losses and clean instances. Intuitively, when labels are correct, small-loss instances are more likely to be the ones which are correctly labeled. Thus, if we train our classifier only using small-loss instances in each mini-bach data, it should be resistant to noisy labels.

However, the above requires that the classifier is reliable enough so that the small-loss instances are indeed clean. The "memorization" effect of deep networks can exactly help us address this problem [2]. Namely, on noisy data sets, even with the existence of noisy labels, deep networks will learn clean and easy pattern in the initial epochs [45, 2]. So, they have the ability to filter out noisy instances using their loss values at the beginning of training. Yet, the problem is that when the number of epochs goes large, they will eventually overfit on noisy labels. To rectify this problem, we want to keep more instances in the mini-batch at the start, i.e., $R(T)$ is large. Then, we gradually increase the drop rate, i.e., $R(T)$ becomes smaller, so that we can keep clean instances and drop those noisy ones before our networks memorize them (details of $R(T)$ will be discussed in Section 4.2).

Based on this idea, we can just use one network in Algorithm 1, and let the classifier evolve by itself. This process is similar to boosting [11] and active learning [7]. However, it is commonly known that boosting and active learning are sensitive to outliers and noise, and a few wrongly selected instances can deteriorate the learning performance of the whole model [12, 3]. This connects with our *second question*, where two classifiers can help.

Intuitively, different classifiers can generate different decision boundaries and then have different abilities to learn. Thus, when training on noisy labels, we also expect that they can have different abilities to filter out the label noise. This motivates us to exchange the selected small-loss instances, i.e., update parameters in $f$ (resp. $g$) using mini-batch instances selected from $g$ (resp. $f$). This process is similar to Co-training [5], and these two networks will adaptively correct the training error by the peer network if the selected instances are not fully clean. Take "peer-review" as a supportive example. When students check their own exam papers, it is hard for them to find any error or bug because they have some personal bias for the answers. Luckily, they can ask peer classmates to review their papers. Then, it becomes much easier for them to find their potential faults. To sum up, as the error from one network will not be directly transferred back itself, we can expect that our Co-teaching method can deal with heavier noise compared with the self-evolving one.

**Relations to Co-training.** Although Co-teaching is motivated by Co-training, the only similarity is that two classifiers are trained. There are fundamental differences between them. (i). Co-training needs two views (two independent sets of features), while Co-teaching needs a single view. (ii) Co-training does not exploit the memorization of deep neural networks, while Co-teaching does. (iii) Co-training is designed for *semi-supervised learning* (SSL), and Co-teaching is for *learning with noisy labels* (LNL); as LNL is not a special case of SSL, we cannot simply translate Co-training from one problem setting to another problem setting.

## 4 Experiments

**Datasets.** We verify the effectiveness of our approach on three benchmark datasets. *MNIST*, *CIFAR-10* and *CIFAR-100* are used here (Table 1), because these data sets are popularly used for evaluation of noisy labels in the literature [13, 31, 33].

Table 1: Summary of data sets used in the experiments.

|  | # of training | # of testing | # of class | image size |
|---|---|---|---|---|
| *MNIST* | 60,000 | 10,000 | 10 | 28×28 |
| *CIFAR-10* | 50,000 | 10,000 | 10 | 32×32 |
| *CIFAR-100* | 50,000 | 10,000 | 100 | 32×32 |

Since all datasets are clean, following [31, 33], we need to corrupt these datasets manually by the noise transition matrix $Q$, where $Q_{ij} = \Pr(\tilde{y} = j|y = i)$ given that noisy $\tilde{y}$ is flipped from clean $y$. Assume that the matrix $Q$ has two representative structures (Figure 2): (1) Symmetry flipping [39]; (2) Pair flipping: a simulation of fine-grained classification with noisy labels, where labelers may make mistakes only within very similar classes. Their precise definition is in Appendix A.

Since this paper mainly focuses on the robustness of our Co-teaching on *extremely* noisy supervision, the noise rate $\epsilon$ is chosen from $\{0.45, 0.5\}$. Intuitively, this means almost half of the instances have noisy labels. Note that, the noise rate $> 50\%$ for pair flipping means over half of the training data

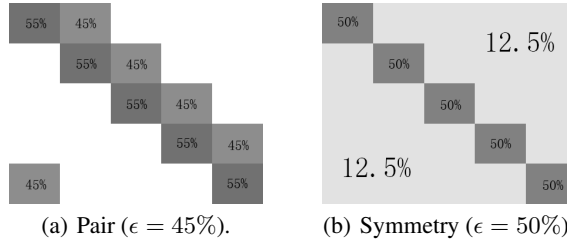

(a) Pair ($\epsilon = 45\%$).      (b) Symmetry ($\epsilon = 50\%$).

Figure 2: Transition matrices of different noise types (using 5 classes as an example).

Table 2: Comparison of state-of-the-art techniques with our Co-teaching approach. In the first column, "large noise": can deal with a large number of classes; "heavy noise": can combat with the heavy noise, i.e., high noise ratio; "flexibility": need not combine with specific network architecture; "no pre-train": can be trained from scratch.

|  | Bootstrap | S-model | F-correction | Decoupling | MentorNet | Co-teaching |
|---|---|---|---|---|---|---|
| large class | ✗ | ✗ | ✗ | ✓ | ✓ | ✓ |
| heavy noise | ✗ | ✗ | ✗ | ✗ | ✓ | ✓ |
| flexibility | ✗ | ✗ | ✓ | ✓ | ✓ | ✓ |
| no pre-train | ✓ | ✗ | ✗ | ✗ | ✓ | ✓ |

have wrong labels that cannot be learned without additional assumptions. As a side product, we also verify the robustness of Co-teaching on *low-level* noisy supervision, where $\epsilon$ is set to $0.2$. Note that pair case is much harder than symmetry case. In Figure 2(a), the true class only has $10\%$ more correct instances over wrong ones. However, the true has $37.5\%$ more correct instances in Figure 2(b).

**Baselines.** We compare the Co-teaching (Algorithm 1) with following state-of-art approaches: (i). Bootstrap [33], which uses a weighted combination of predicted and original labels as the correct labels, and then does back propagation. Hard labels are used as they yield better performance; (ii). S-model [13], which uses an additional softmax layer to model the noise transition matrix; (iii). F-correction [31], which corrects the prediction by the noise transition matrix. As suggested by the authors, we first train a standard network to estimate the transition matrix; (iv). Decoupling [26], which updates the parameters only using the samples which have different prediction from two classifiers; and (v). MentorNet [17]. An extra teacher network is pre-trained and then used to filter out noisy instances for its student network to learn robustly under noisy labels. Then, student network is used for classification. We used self-paced MentorNet in this paper. (vi). As a baseline, we compare Co-teaching with the standard deep networks trained on noisy datasets (abbreviated as Standard). Above methods are systematically compared in Table 2. As can be seen, our Co-teaching method does not rely on any specific network architectures, which can also deal with a large number of classes and is more robust to noise. Besides, it can be trained from scratch. These make our Co-teaching more appealing for practical usage. Our implementation of Co-teaching is available at https://github.com/bhanML/Co-teaching.

**Network structure and optimizer.** For the fair comparison, we implement all methods with default parameters by PyTorch, and conduct all the experiments on a NIVIDIA K80 GPU. CNN is used with Leaky-ReLU (LReLU) active function [25], and the detailed architecture is in Table 3. Namely, the 9-layer CNN architecture in our paper follows "Temporal Ensembling" [21] and "Virtual Adversarial Training" [29], since the network structure we used here is standard test bed for weakly-supervised learning. For all experiments, Adam optimizer (momentum=0.9) is with an initial learning rate of 0.001, and the batch size is set to 128 and we run 200 epochs. Besides, dropout and batch-normalization are also used. As deep networks are highly nonconvex, even with the same network and optimization method, different initializations can lead to different local optimal. Thus, following [26], we also take two networks with the same architecture but different initializations as two classifiers.

**Experimental setup.** Here, we assume the noise level $\epsilon$ is known and set $R(T) = 1 - \tau \cdot \min(T/T_k, 1)$ with $T_k = 10$ and $\tau = \epsilon$. If $\epsilon$ is not known in advanced, $\epsilon$ can be inferred using validation sets [23, 43]. The choices of $R(T)$ and $\tau$ are analyzed in Section 4.2. Note that $R(T)$ only depends on the memorization effect of deep networks but not any specific datasets.

As for performance measurements, first, we use the test accuracy, i.e., *test Accuracy = (# of correct predictions) / (# of test dataset)*. Besides, we also use the label precision in each mini-batch, i.e., *label Precision = (# of clean labels) / (# of all selected labels)*. Specifically, we sample $R(T)$ of

Table 3: CNN models used in our experiments on *MNIST*, *CIFAR-10*, and *CIFAR-100*. The slopes of all LReLU functions in the networks are set to 0.01.

| CNN on *MNIST* | CNN on *CIFAR-10* | CNN on *CIFAR-100* |
|---|---|---|
| 28×28 Gray Image | 32×32 RGB Image | 32×32 RGB Image |
| | 3×3 conv, 128 LReLU | |
| | 3×3 conv, 128 LReLU | |
| | 3×3 conv, 128 LReLU | |
| | 2×2 max-pool, stride 2 | |
| | dropout, $p = 0.25$ | |
| | 3×3 conv, 256 LReLU | |
| | 3×3 conv, 256 LReLU | |
| | 3×3 conv, 256 LReLU | |
| | 2×2 max-pool, stride 2 | |
| | dropout, $p = 0.25$ | |
| | 3×3 conv, 512 LReLU | |
| | 3×3 conv, 256 LReLU | |
| | 3×3 conv, 128 LReLU | |
| | avg-pool | |
| dense 128→10 | dense 128→10 | dense 128→100 |

Table 4: Average test accuracy on *MNIST* over the last ten epochs.

| Flipping-Rate | Standard | Bootstrap | S-model | F-correction | Decoupling | MentorNet | Co-teaching |
|---|---|---|---|---|---|---|---|
| Pair-45% | 56.52% | 57.23% | 56.88% | 0.24% | 58.03% | 80.88% | **87.63%** |
| | ±0.55% | ±0.73% | ±0.32% | ±0.03% | ±0.07% | ±4.45% | ±0.21% |
| Symmetry-50% | 66.05% | 67.55% | 62.29% | 79.61% | 81.15% | 90.05% | **91.32%** |
| | ±0.61% | ±0.53% | ±0.46% | ±1.96% | ±0.03% | ±0.30% | ±0.06% |
| Symmetry-20% | 94.05% | 94.40% | 98.31% | **98.80%** | 95.70% | 96.70% | 97.25% |
| | ±0.16% | ±0.26% | ±0.11% | ±0.12% | ±0.02% | ±0.22% | ±0.03% |

small-loss instances in each mini-batch, and then calculate the ratio of clean labels in the small-loss instances. Intuitively, higher label precision means less noisy instances in the mini-batch after sample selection, and the algorithm with higher label precision is also more robust to the label noise. All experiments are repeated five times. The error bar for STD in each figure has been highlighted as a shade. Besides, the full Y-axis versions for all figures are in Appendix B.

## 4.1 Comparison with the State-of-the-Arts

**Results on *MNIST*.** Table 4 reports the accuracy on the testing set. As can be seen, on the symmetry case with 20% noisy rate, which is also the easiest case, all methods work well. Even Standard can achieve 94.05% test set accuracy. Then, when noisy rate raises to 50%, Standard, Bootstrap, S-model and F-correction fail, and their accuracy decrease lower than 80%. Methods based on "selected instances", i.e., Decoupling, MentorNet and Co-teaching are better. Among them, Co-teaching is the best. Finally, in the hardest case, i.e., pair case with 45% noisy rate, Standard, Bootstrap and S-Model cannot learn anything. Their testing accuracy keep the same as the percentage of clean instances in the training dataset. F-correct fails totally, and it heavily relies on the correct estimation of the underneath transition matrix. Thus, when Standard works, it can work better than Standard; then, when Standard fails, it works much worse than Standard. In this case, our Co-teaching is again the best, which is also much better than the second method, i.e. 87.53% for Co-teaching vs. 80.88% for MentorNet.

In Figure 3 , we show test accuracy vs. number of epochs. In all three plots, we can clearly see the memorization effects of networks, i.e., test accuracy of Standard first reaches a very high level and then gradually decreases. Thus, a good robust training method should stop or alleviate the decreasing processing. On this point, all methods except Bootstrap work well in the easiest Symmetry-20% case. However, only MentorNet and our Co-teaching can combat with the other two harder cases, i.e., Pair-45% and Symmetry-50%. Besides, our Co-teaching consistently achieves higher accuracy than MentorNet, and is the best method in these two cases.

To explain such good performance, we plot label precision vs. number of epochs in Figure 4. Only MentorNet, Decoupling and Co-teaching are considered here, as they are methods do instance selection during training. First, we can see Decoupling fails to pick up clean instances, and its label precision is the same as Standard which does not compact with noisy label at all. The reason is that Decoupling does not utilize the memorization effects during training. Then, we can see Co-teaching and MentorNet can successfully pick clean instances out. These two methods tie on the easier

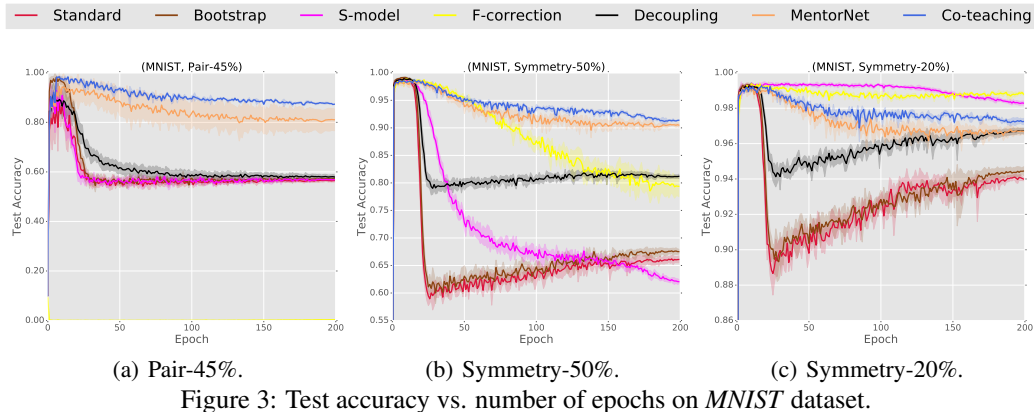

(a) Pair-45%.  (b) Symmetry-50%.  (c) Symmetry-20%.

Figure 3: Test accuracy vs. number of epochs on *MNIST* dataset.

Symmetry-50% and Symmetry-20%, when our Co-teaching achieve higher precision on the hardest Pair-45% case. This shows our approach is better at finding clean instances.

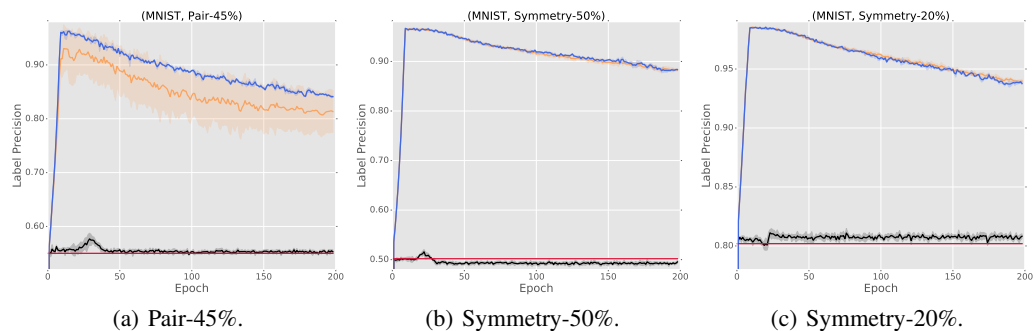

(a) Pair-45%.  (b) Symmetry-50%.  (c) Symmetry-20%.

Figure 4: Label precision vs. number of epochs on *MNIST* dataset.

Finally, note that while in Figure 4(b) and (c), MentorNet and Co-teaching tie together. Co-teaching still gets higher testing accuracy (Table 4). Recall that MentorNet is a self-evolving method, which only uses one classifier, while Co-teaching uses two. The better accuracy comes from the fact Co-teaching further takes the advantage of different learning abilities of two classifiers.

**Results on *CIFAR-10*.** Test accuracy is shown in Table 5. As we can see, the observations here are consistently the same as these for *MNIST* dataset. In the easiest Symmetry-20% case, all methods work well. F-correction is the best, and our Co-teaching is comparable with F-correction. Then, all methods, except MentorNet and Co-teaching, fail on harder, i.e., Pair-45% and Symmetry-50% cases. Between these two, Co-teaching is the best. In the extreme Pair-45% case, Co-teaching is at least 14% higher than MentorNet in test accuracy.

Table 5: Average test accuracy on *CIFAR-10* over the last ten epochs.

| Flipping,Rate | Standard | Bootstrap | S-model | F-correction | Decoupling | MentorNet | Co-teaching |
|---|---|---|---|---|---|---|---|
| Pair-45% | 49.50% | 50.05% | 48.21% | 6.61% | 48.80% | 58.14% | **72.62%** |
| | ±0.42% | ±0.30% | ±0.55% | ±1.12% | ±0.04% | ±0.38% | ±0.15% |
| Symmetry-50% | 48.87% | 50.66% | 46.15% | 59.83% | 51.49% | 71.10% | **74.02%** |
| | ±0.52% | ±0.56% | ±0.76% | ±0.17% | ±0.08% | ±0.48% | ±0.04% |
| Symmetry-20% | 76.25% | 77.01% | 76.84% | **84.55%** | 80.44% | 80.76% | 82.32% |
| | ±0.28% | ±0.29% | ±0.66% | ±0.16% | ±0.05% | ±0.36% | ±0.07% |

Figure 5 shows test accuracy and label precision vs. number of epochs. Again, on test accuracy, we can see Co-teaching strongly hinders neural networks from memorizing noisy labels. Thus, it works much better on the harder Pair-45% and Symmetry-50% cases. On label precision, while Decoupling fails to find clean instances, both MentorNet and Co-teaching can do this. However, due to the usage of two classifiers, Co-teaching is stronger.

**Results on *CIFAR-100*.** Finally, we show our results on *CIFAR-100*. The test accuracy is in Table 6. Test accuracy and label precision vs. number of epochs are in Figure 6. Note that there are only 10 classes in *MNIST* and *CIFAR-10* datasets. Thus, overall the accuracy is much lower than previous

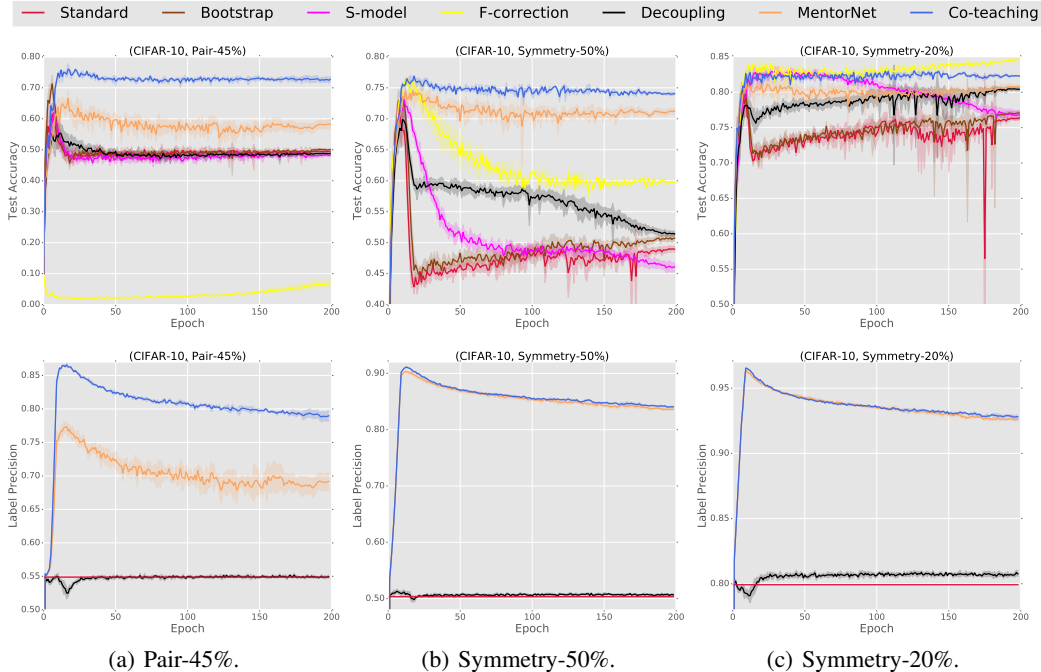

Figure 5: Results on *CIFAR-10* dataset. Top: test accuracy vs. number of epochs; bottom: label precision vs. number of epochs.

ones in Tables 4 and 5. However, the observations are the same as previous datasets. We can clearly see our Co-teaching is the best on harder and noisy cases.

Table 6: Average test accuracy on *CIFAR-100* over the last ten epochs.

| Flipping,Rate | Standard | Bootstrap | S-model | F-correction | Decoupling | MentorNet | Co-teaching |
|---|---|---|---|---|---|---|---|
| Pair-45% | 31.99% | 32.07% | 21.79% | 1.60% | 26.05% | 31.60% | **34.81%** |
| | ±0.64% | ±0.30% | ±0.86% | ±0.04% | ±0.03% | ±0.51% | ±0.07% |
| Symmetry-50% | 25.21% | 21.98% | 18.93% | 41.04% | 25.80% | 39.00% | **41.37%** |
| | ±0.64% | ±6.36% | ±0.39% | ±0.07% | ±0.04% | ±1.00% | ±0.08% |
| Symmetry-20% | 47.55% | 47.00% | 41.51% | **61.87%** | 44.52% | 52.13% | 54.23% |
| | ±0.47% | ±0.54% | ±0.60% | ±0.21% | ±0.04% | ±0.40% | ±0.08% |

## 4.2 Choices of $R(T)$ and $\tau$

Deep networks initially fit clean (easy) instances, and then fit noisy (hard) instances progressively. Thus, intuitively $R(T)$ should meet following requirements: (i). $R(T) \in [\tau, 1]$, where $\tau$ depends on the noise rate $\epsilon$; (ii). $R(1) = 1$, which means we do not need to drop any instances at the beginning. At the initial learning epochs, we can safely update the parameters of deep neural networks using entire noisy data, because the networks will not memorize the noisy data at the early stage [2]; (iii). $R(T)$ should be a non-increasing function on $T$, which means that we need to drop more instances when the number of epochs gets large. This is because as the learning proceeds, the networks will eventually try to fit noisy data (which tends to have larger losses compared to clean data). Thus, we need to ignore them by not updating the networks parameters using large loss instances [2]. The *MNIST* dataset is used in the sequel.

Based on above principles, to show how the decay of $R(T)$ affects Co-teaching, first, we let $R(T) = 1 - \tau \cdot \min\{T^c/T_k, 1\}$ with $\tau = \epsilon$, where three choices of $c$ should be considered, i.e., $c = \{0.5, 1, 2\}$. Then, three values of $T_k$ are considered, i.e., $T_k = \{5, 10, 15\}$. Results are in Table 7. As can be seen, the test accuracy is stable on the choices of $T_k$ and $c$ here. The previous setup ($c = 1$ and $T_k = 10$) works well but does not lead to the best performance. To show the impact of $\tau$, we vary $\tau = \{0.5, 0.75, 1, 1.25, 1.5\}\epsilon$. Note that, $\tau$ cannot be zero. In this case, no gradient will be back-propagated and the optimization will stop. Test accuracy is in Table 8. We can see, with more dropped instances, the performance can be improved. However, if too many instances are dropped, networks may not get sufficient training data and the performance can deteriorate. We set $\tau = \epsilon$ in Section 4.1, and it works well but not necessarily leads to the best performance.

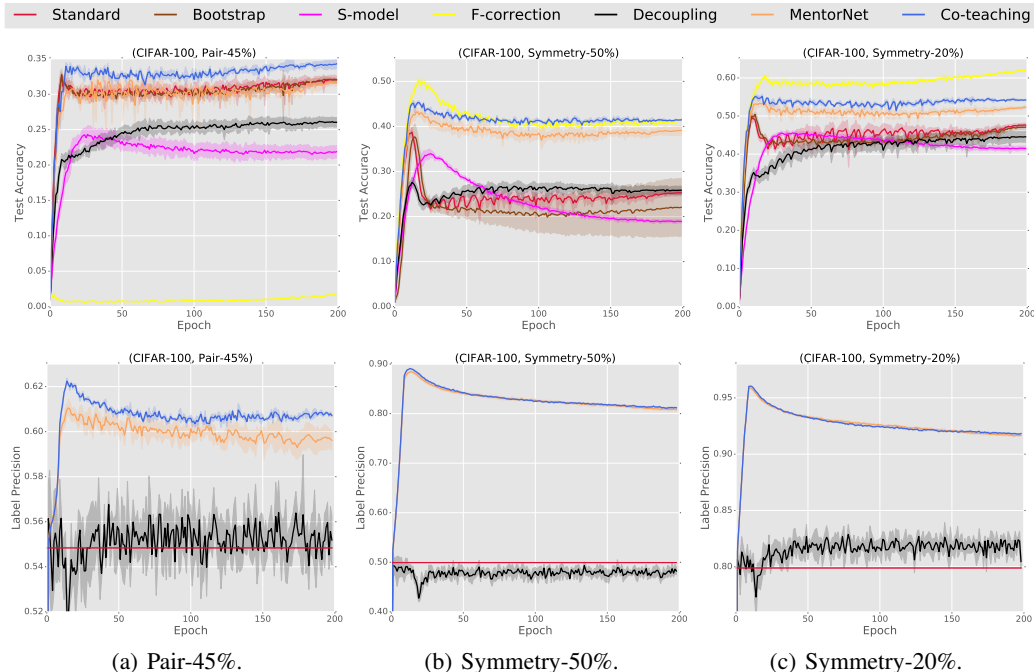

(a) Pair-45%.  (b) Symmetry-50%.  (c) Symmetry-20%.

Figure 6: Results on *CIFAR-100* dataset. Top: test accuracy vs. number of epochs; bottom: label precision vs. number of epochs.

Table 7: Average test accuracy on *MNIST* over the last ten epochs.

|  |  | $c = 0.5$ | $c = 1$ | $c = 2$ |
|---|---|---|---|---|
| Pair-45% | $T_k = 5$ | 75.56%±0.33% | 87.59%±0.26% | 87.54%±0.23% |
|  | $T_k = 10$ | **88.43%±0.25%** | 87.56%±0.12% | 87.93%±0.21% |
|  | $T_k = 15$ | **88.37%±0.09%** | 87.29%±0.15% | **88.09%±0.17%** |
| Symmetry-50% | $T_k = 5$ | 91.75%±0.13% | 91.75%±0.12% | **92.20%±0.14%** |
|  | $T_k = 10$ | 91.70%±0.21% | 91.55%±0.08% | 91.27%±0.13% |
|  | $T_k = 15$ | 91.74%±0.14% | 91.20%±0.11% | 91.38%±0.08% |
| Symmetry-20% | $T_k = 5$ | 97.05%±0.06% | 97.10%±0.06% | 97.41%±0.08% |
|  | $T_k = 10$ | 97.33%±0.05% | 96.97%±0.07% | **97.48%±0.08%** |
|  | $T_k = 15$ | 97.41%±0.06% | 97.25%±0.09% | **97.51%±0.05%** |

Table 8: Average test accuracy of Co-teaching with different $\tau$ on *MNIST* over the last ten epochs.

| Flipping,Rate | $0.5\epsilon$ | $0.75\epsilon$ | $\epsilon$ | $1.25\epsilon$ | $1.5\epsilon$ |
|---|---|---|---|---|---|
| Pair-45% | 66.74%±0.28% | 77.86%±0.47% | 87.63%±0.21% | **97.89%±0.06%** | 69.47%±0.02% |
| Symmetry-50% | 75.89%±0.21% | 82.00%±0.28% | 91.32%±0.06% | **98.62%±0.05%** | 79.43%±0.02% |
| Symmetry-20% | 94.94%±0.09% | 96.25%±0.06% | 97.25%±0.03% | 98.90%±0.03% | **99.39%±0.02%** |

## 5 Conclusion

This paper presents a simple but effective learning paradigm called Co-teaching, which trains deep neural networks robustly under noisy supervision. Our key idea is to maintain two networks simultaneously, and cross-trains on instances screened by the "small loss" criteria. We conduct simulated experiments to demonstrate that, our proposed Co-teaching can train deep models robustly with the extremely noisy supervision. In future, we can extend our work in the following aspects. First, we can adapt Co-teaching paradigm to train deep models under other weak supervisions, e.g., positive and unlabeled data [19]. Second, we would investigate the theoretical guarantees for Co-teaching. Previous theories for Co-training are very hard to transfer into Co-teaching, since our setting is fundamentally different. Besides, there is no analysis for generalization performance on deep learning with noisy labels. Thus, we leave the generalization analysis as a future work.

**Acknowledgments.**

MS was supported by JST CREST JPMJCR1403. IWT was supported by ARC FT130100746, DP180100106 and LP150100671. BH would like to thank the financial support from RIKEN-AIP. XRY was supported by NSFC Project No. 61671481. QY would give special thanks to Weiwei Tu and Yuqiang Chen from 4Paradigm Inc. We gratefully acknowledge the support of NVIDIA Corporation with the donation of the Titan Xp GPU used for this research.

## Footnotes

*The first two authors (Bo Han and Quanming Yao) made equal contributions. The implementation is available at https://github.com/bhanML/Co-teaching.

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
