[Supplementary Material · 5143_supp.pdf]

# A  Definition of noise

The definition of transition matrix $Q$ is as follow. $n$ is number of the class.

$$\text{Pair flipping:} \quad Q = \begin{bmatrix} 1-\epsilon & \epsilon & 0 & \dots & 0 \\ 0 & 1-\epsilon & \epsilon & & 0 \\ \vdots & & \ddots & \ddots & \vdots \\ 0 & & & 1-\epsilon & \epsilon \\ \epsilon & 0 & \dots & 0 & 1-\epsilon \end{bmatrix},$$

$$\text{Symmetry flipping:} \quad Q = \begin{bmatrix} 1-\epsilon & \frac{\epsilon}{n-1} & \dots & \frac{\epsilon}{n-1} & \frac{\epsilon}{n-1} \\ \frac{\epsilon}{n-1} & 1-\epsilon & \frac{\epsilon}{n-1} & \dots & \frac{\epsilon}{n-1} \\ \vdots & & \ddots & & \vdots \\ \frac{\epsilon}{n-1} & \dots & \frac{\epsilon}{n-1} & 1-\epsilon & \frac{\epsilon}{n-1} \\ \frac{\epsilon}{n-1} & \frac{\epsilon}{n-1} & \dots & \frac{\epsilon}{n-1} & 1-\epsilon \end{bmatrix}.$$

# B  Full Y-axis figures

## B.1  *MNIST*

(a) Pair-45%.  (b) Symmetry-50%.  (c) Symmetry-20%.

Figure 7: Results on *MNIST* dataset. Top: test accuracy vs. number of epochs; bottom: label precision vs. number of epochs.

## B.2  *CIFAR-10*

Figure 8: Results on *CIFAR-10* dataset. Top: test accuracy vs. number of epochs; bottom: label precision vs. number of epochs.

## B.3  *CIFAR-100*

Figure 9: Results on *CIFAR-100* dataset. Top: test accuracy vs. number of epochs; bottom: label precision vs. number of epochs.