[Reviews · NeurIPS 2018]

Reviewer 1



This paper proposes and empirically investigates a co-teaching learning strategy that makes use of two networks to cross-train data with noisy labels for robust learning. Experiments on MNIST, CIFAR10 and CIFAR100 seem to be supportive. The paper is well structured and well written, but technically I have following concerns. 1. Since the proposed co-teaching strategy is motivated by co-training, one can tell the similarity between the two. Typically, in a semi-supervised learning setup by co-training, the labeled and unlabeled data is known. It starts with the labeled data and improves two models based on the confidence of the unlabeled data. In the proposed co-teaching setup, however, whether a label is clean or noisy is unknown. So co-teaching starts with some "clean" labels and gradually builds two good models to deal with the noisy labels. Otherwise, co-teaching works very similar to co-training. 2. Since the quality of the labels is unknown, co-teaching has to estimate a few variables, for instance, the confidence of the labels and the noise rate. The former is done by so-called "small-loss", which is argued to give rise to "support vectors" and the latter is simply assumed to be known ($\tau$) in the experiments. This is one of my concerns. First of all, it is not clear how the estimate of $\tau$ would impact the performance of co-teaching. It would be helpful to show some results. Second, there is an implicit assumption behind co-teaching. That is the clean and noisy labels are uniformly distributed across mini-batches. In practice, both assumptions may not hold. Therefore, it is important to show the results with estimated $\tau$. 3. "dealing with extremely noisy labels" is claimed to be the focus of this paper. However, it is not clear what noisy rate is considered "extremely noisy". It may be a reasonable speculation that noise rate >50% is extremely noisy. Furthermore, why does co-teaching only work well for extremely noisy labels while not for slightly noisy labels? Wouldn't it be reasonable to show its behavior through a spectrum of noise rate, say, from clean, 10%, 20%, all the way to 80% (in a controlled manner, for instance, only on one dataset)? 4. Co-training has a very good theory backing it. The proposed co-teaching in this paper is basically empirical observations. It would be nice to come up with some similar analysis as co-training. Overall, I find this paper clearly written. The idea is interesting but significance of the reported work is not overwhelming. There are some fundamental issues regarding the approach and the experimental results that need to be clarified.

Reviewer 2



The paper proposes co-teaching which uses the result that deep learning models learn easy instances first. One network is used to select the examples for the target network. The paper shows that for two type of noise, Co-teaching outperforms several baselines including Mentornet. The authors have demonstrated the effectiveness on vision and language datasets. I have one suggestion: The paper is mostly empirical and analysis will strengthen the results.

Reviewer 3



This paper is about a novel method to robustly train deep neural networks when datasets contains extreme noise in the labels. The idea is quite simple and elegant. Based on the fact that neural networks have the "memorization effect" (i.e. the ability to predict correctly to some extend before being losing accuracy due to the memorisation of the noise), the approach exploits it to teach two neural networks concurrently by selecting only R(T) samples on each mini batch, which have a small loss and are probably well classified. The choice of selecting only R(T) istances is well motivated, as it is the need for using two networks. The idea of using two networks is similar in spirit to the classic co-training and has connection to the Decoupling approach [19] but it is applied to the task of training networks with noisy labels. Experiments are performed on MNIST, CIFAR10, CIFAR100 and show that the approach is superior to several state of the art approaches. Strengths: + Real data is typically noisy, so this line of research is of high interest and related to many applications of deep neural networks. + The paper is very well written, very ease to read, complete. I found it to be very interesting. + The method works in general better than the compared state of the art works, especially when high level of noise is present. + Experiments are consistent and well performed. Weakness: - It would have been interesting a test with a bigger, state of the art network for classification on ImageNet dataset. Even AlexNet would have been interesting to see. - The hypothesis that the noise parameter tau is fixed and known is limited. It would have been interesting to have at least an experiment where the value is tested with "wrong" values such as too low or too high. Easy fixable issues: - Table 1: CIFAR100 has 100 classes, not 1k. - Table 2: "large noise" -> "large class" - row 139: NIVIDIA -> NVIDIA - row 183: 87.53 % when in table 3 is 87.63 %. - row 189: pair 20% while Fig 3 (a) is pair 45%. - row 189: "higher accuracy MentorNet" -> "higher accuracy than MentorNet" - row 192-3: as they are methods *that* do instance *selection* during training.